# Pseudo-Probability Unlearning: Towards efficient and privacy-preserving machine unlearning

## Abstract

Machine unlearning—enabling a trained model to forget specific data—is crucial for addressing biased data and adhering to privacy regulations like the General Data Protection Regulation (GDPR)'s "right to be forgotten." Recent works have paid little attention to privacy concerns, leaving the data intended for forgetting vulnerable to membership inference attacks. Moreover, they often come with high computational overhead. In this work, we propose Pseudo-Probability Unlearning (PPU), a novel method that enables models to forget data efficiently and in a privacy-preserving manner. Our method replaces the final-layer output probabilities of the neural network with pseudo-probabilities for the data to be forgotten. These pseudo-probabilities follow either a uniform distribution or align with the model's overall distribution, enhancing privacy and reducing risk of membership inference attacks. Our optimization strategy further refines the predictive probability distributions and updates the model's weights accordingly, ensuring effective forgetting with minimal impact on the model's overall performance. Through comprehensive experiments on multiple benchmarks, our method achieves over 20% improvements in forgetting error compared to the state-of-the-art. Additionally, our method enhances privacy by preventing the forgotten set from being inferred to around random guesses.

## 1 Introduction

Machine unlearning, which focuses on eliminating the negative impact of specific data subsets—such as biased, erroneous, or privacy-leaking instances (Jagielski et al., 2018; Yang et al., 2024)—used in model training (Baumhauer et al., 2022; Fu et al., 2022; Golatkar et al., 2020a;b; Guo et al., 2019; Kim & Woo, 2022; Mehta et al., 2022; Nguyen et al., 2020; Shah et al., 2023), has emerged as a critical area of research. Its significance is increasing due to growing concerns about data privacy (Pardau, 2018), legal requirements for data deletion(Mantelero, 2013), and the necessity for models to adapt to new information without complete retraining. Though the most straightforward approach is to retrain the model with a new dataset that excludes the data needing removal, this approach is computationally expensive and needs continuous access to the training set.

There are two main challenges in existing machine unlearning methods. On the one hand, they still face high computational time (Xu et al., 2023) without retraining the model if they aim to maintain decent **unlearning performance** in two aspects: the efficiency of forgetting the specified data subset, and the need to maintain performance on the remaining data. For example, an existing machine unlearning work (Kurmanji et al., 2024) requires 4.30 seconds to forget 25 data samples, while retraining takes 4.21 seconds. The retraining process is even faster due to the unlearning method's complexity of the loss computation.

On the other hand, existing methods are vulnerable to privacy leakage attacks (Hu et al., 2024), where an attacker can infer which data is within the forgetting set from the post-unlearning models. This still violates the right to be forgotten, even though the model has been updated to remove the data. This vulnerability arises because existing methods typically require the model to perform poorly (i.e., have a high loss) on the forgetting set, making it easier to distinguish from the retraining set. We denote this vulnerability as **privacy leakage**, measured by Membership Inference Attacks

(MIA) (Hui et al., 2021). There is a lack of existing work addressing privacy leakage in machine unlearning; to the best of our knowledge, Kurmanji et al. (Kurmanji et al., 2024) has considered this issue, but it is computationally costly as mentioned earlier.

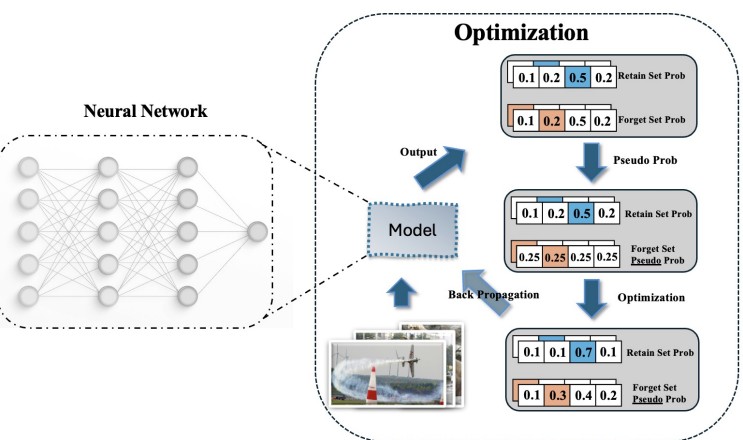

Figure 1: This is an overview of Pseudo-Probability Unlearning (PPU). In this approach, we extract the output layer probabilities and replace the forget set probabilities with pseudo-probabilities. After performing optimization, the model's weights are fine-tuned using the refined pseudo-probabilities.

To address these issues, we propose Pseudo-Probability Unlearning (PPU) in Figure 1, which targets the final-layer output probabilities of the model and replaces them with pseudo-probabilities for the data to be forgotten, thus being computationally efficient. To achieve good forgetting performance, these pseudo-probabilities for the forgetting set are initialized from a uniform distribution and are further refined to maintain performance on the remaining data. To protect privacy, the entire model's weights are updated with the objective of ensuring that the pseudo-probabilities do not deviate too far from the original model's output probabilities, thus making the forgetting set indistinguishable from the remaining data. Besides, we provide a proof for the efficiency of Pseudo-Probability, making it theoretically sound. Extensive evaluations show that PPU reduces computational time by half compared to existing methods while improving unlearning performance and preventing privacy leakage, reducing the success rate of membership inference attacks to around random guessing.

## 2 RELATED WORK

The discussion on unlearning has been broadened to include two principle paradigms: exact unlearning and approximate unlearning (Izzo et al., 2021). Exact unlearning mandates that the performance of a model, post-unlearning, should be indistinguishable from that of a model retrained in the absence of the forgotten data. In this vein, Brophy and Lowd (Brophy & Lowd, 2021), along with Schelter et al. (Schelter et al., 2021), applied exact unlearning methods specifically to random forest models. Similarly, Ginart et al. (Ginart et al., 2019) developed an exact unlearning technique for k-means clustering. Despite the efficacy and precision of exact unlearning approaches in diminishing the influence of specific data, they face significant constraints related to underlying assumptions and scalability issues, as highlighted by Xu et al. (Xu et al., 2023). In particular, these methods are unsuitable for models such as Convolutional Neural Networks (CNN) (O'Shea & Nash, 2015) and Residual Networks (ResNet) (He et al., 2016). To address this, Golatkar et al. (Golatkar et al., 2020a) introduced the concept of selective unlearning, aiming to achieve forgetting by adjusting model weights. Moreover, Bourtoule et al. (Bourtoule et al., 2021) introduced the Sharded, Isolated, Sliced, and Aggregated (SISA) training approach, which ingeniously reduces the influence of individual data points on the training process. However, this approach can significantly compromise the model's performance and generalization capacity, especially when multiple data points need to be unlearned. Furthermore, Golatkar et al. (Golatkar et al., 2020b) proposed approximating the weights

that would result from unlearning by using a linearization inspired by Neural Tangent Kernel (NTK) theory (Jacot et al., 2018).

Approximate unlearning is designed to diminish the impact of data designated for removal to a tolerable extent, rather than achieving its complete elimination. This method recognizes the inherent difficulties in fully erasing the influence of data from complex models. Cao and Yang (Cao & Yang, 2015) developed a strategy to reconfigure learning algorithms to ease data deletion, but this method faces scalability issues with more complex models. Wu et al. (Wu et al., 2020) proposed using cached information from the original training to ease the retraining process, though this technique struggles with large-scale data removal. Taking a different approach, Kurmanji et al. (Kurmanji et al., 2024) introduced a novel approach involving the optimization of the min-max problem to improve the unlearning process. Zhang et al. (Zhang et al., 2022) also presented a method that leverages quantized gradients and randomized smoothing to potentially prevent the need for future unlearning, offering certain guarantees under specific conditions. Nevertheless, a data deletion request that significantly alters the data distribution, such as class unlearning, may exceed the "deletion budget" and challenge the assumptions underlying their approach.

Overall, most approaches improve upon retraining but still require significant computational resources, making them less practical for large-scale applications. Although some approximations aim to enhance efficiency, few explicitly prioritize privacy, which remains a critical concern. Additionally, there is still room to reduce forget set error and further improve privacy protections, highlighting the need for more balanced and effective solutions.

## 3  Notations and Problem Definition

Consider a dataset $\mathcal{D} = (\mathbf{x}_i, y_i)_{i=1}^{N}$, composed of $N$ data points, where each instance consists of an input feature vector $\mathbf{x}_i$ and its corresponding label $y_i$. Let $f(\cdot; \mathbf{w})$ represent a function implemented by a deep neural network, parameterized by the weights $\mathbf{w}$. In this context, we are provided with a "forget set" $\mathcal{D}_{\text{fog}} = (\mathbf{x}_f, y_f)_{f=1}^{N_f} \subset \mathcal{D}$, consisting of $N_f$ instances extracted from $\mathcal{D}$, as well as a "retain set" $\mathcal{D}_{\text{ret}}(\mathbf{x}_r, y_r)_{r=1}^{N_r} \subset \mathcal{D}$ containing $N_r$ training samples. For simplicity, we assume that $\mathcal{D}_{\text{ret}}$ is the complement of $\mathcal{D}_{\text{fog}}$, satisfying the condition $\mathcal{D}_{\text{fog}} \cup \mathcal{D}_{\text{ret}} = \mathcal{D}$ and $N_f + N_r = N$, thereby covering the entire original dataset.

This formulation sets the foundation for exploring methods capable of effectively "unlearning" the specified $\mathcal{D}_{\text{fog}}$ from the original model, ensuring that the resulting model's performance is primarily influenced by the data in $\mathcal{D}_{\text{ret}}$. The goal of deep machine unlearning is to derive a new set of weights, $\mathbf{w}_u$, such that the updated model, $f(\cdot; \mathbf{w}_u)$, effectively "erases" the information related to $\mathcal{D}_{\text{fog}}$. This process should be carried out without compromising the model's utility, as demonstrated by its performance on $\mathcal{D}_{\text{ret}}$ and its ability to generalize to unseen data. We propose an optimization technique that refines the model's last layer output predictive probability distribution to efficiently achieve unlearning objectives.

## 4  Methods

Building on the foundational framework, we propose an approach for deep machine unlearning that leverages pseudo-label optimization. In this method, we construct a matrix where each column represents a class, and each row corresponds to the probability of a data point belonging to each class. If the training dataset contains $N$ data points and $k$ classes, the resulting matrix has dimensions $N \times k$. For instance, in a neural network trained on the CIFAR-10 dataset with 50,000 training images, the matrix would have 50,000 rows and 10 columns. This matrix serves as the foundation for refining the model's predictions and optimizing the unlearning process.

We define the output probabilities as $p$, where each $p$ vector has a length of $k$ for each data point. Specifically, we assign the probabilities for the forgotten data as $p_f$ and for the retained data as $p_r$. Thus, each cell in the matrix represents the probability of a data point belonging to a particular class, consisting of values from both $p_f$ and $p_r$. For a data point $x_1$ in the forget set, the probability of $x_1$ belonging to class $k$ is denoted as $p_{f1}(k)$. Here, we define $f(x; \mathbf{w})$ as the output probability generated by the input $x$ when passed through the model with weights $\mathbf{w}$. Additionally, $f_k(x_f; \mathbf{w})$ represents the probability of the forgotten data point $x_f$ belonging to class $k$.

The core of our method lies in the formulation of an optimization objective tailored to adjust the model's output distribution in such a manner that it effectively "forgets" the information related to the forget set $\mathcal{D}_f$, while maintaining or even enhancing its performance on the retain set $\mathcal{D}_r$. To this end, we introduce an optimization objective designed to measure the discrepancy between the desired output distribution and the one currently produced by the model with original weights $\mathbf{w}$.

## 4.1 PSEUDO-PROBABILITY REFINEMENT FOR DEEP MACHINE UNLEARNING

In the proposed formulation, $\{\hat{p}_{fi}\}_{i=1}^{N_f}$ represents the set of pseudo-probabilities for the forget set. To ensure the model forgets this data, we can either set the pseudo-probabilities to a uniform distribution or generate them randomly. These strategies help to "mask" or obscure the model's previous knowledge about the forget set, making it harder for the model to retain those associations with the forget set, thereby improves the precision and effectiveness of the unlearning process. While the pseudo-probabilities for the forget set $\hat{p}_f$ are adjusted to obscure the model's learned associations with $\mathcal{D}_{\text{fog}}$, the probabilities for the retain set $\hat{p}_r$ are $f(x_r; \mathbf{w})$ to ensure consistency with the model's knowledge of $\mathcal{D}_{\text{ret}}$.

During optimization, $\hat{p}_r$ evolves from the initial model output $f(x_r, \mathbf{w})$ into a distribution that better reflects the knowledge the model should retain. The objective function incorporates a KL-divergence term for both the forget set and the retain set, weighted by the parameter $\lambda$. This ensures a balanced approach to managing knowledge across both sets. The aim is to reach a state of neutrality or ignorance for the forget set, while ensuring the output distribution of the retain set aligns closely with the target distribution. We assume that the total probability for class $k$ across all data points is $M_k$, and we strive to maintain this constant, regardless of any changes in the probabilities. This constraint prevents the unlearning process from excessively distorting the model's ability to forget the specified data while retaining knowledge about the retained set.

$$\min_{\{f(x_f;\mathbf{w})\}_{f=1}^{N_f}, \{f(x_r;\mathbf{w})\}_{r=1}^{N_r}} \left( \sum_{f=1}^{N_f} D_{KL}(f(x_f;\mathbf{w})\|\hat{p}_f) + \lambda \sum_{r=1}^{N_r} D_{KL}(f(x_r;\mathbf{w})\|\hat{p}_r) \right) \quad (1)$$

$$\text{subject to} \quad \sum_{f=1}^{N_f} f_k(x_f;\mathbf{w}) + \sum_{r=1}^{N_r} f_k(x_r;\mathbf{w}) = M_k, \quad \forall k, \quad (2)$$

$$\sum_{k=1}^{K} f_k(x_f;\mathbf{w}) = 1, \quad \forall f, \quad \sum_{k=1}^{K} f_k(x_r;\mathbf{w}) = 1, \quad \forall r, \quad (3)$$

$$f_k(x_f;\mathbf{w}) \in [0,1], \quad \forall f,k, \quad f_k(x_r;\mathbf{w}) \in [0,1], \quad \forall r,k. \quad (4)$$

Additionally, within the constraints, we must ensure that the sum of the probabilities across all classes for each data point equals 1. Furthermore, the probabilities should be bounded between 0 and 1, meaning they must be greater than or equal to 0 and less than or equal to 1.

The constraints guarantee that the pseudo-labels for both sets adhere to predefined distributions and form valid probability distributions over class labels. This optimization strategy thus offers a comprehensive framework for managing the objectives of unlearning and adaptive retention within a machine learning model.

### 4.1.1 CONVERGENCE TO THE UNIQUE OPTIMAL SOLUTION

To address computational efficiency, particularly for large datasets, we adopt an iterative solution reminiscent of coordinate ascent algorithms applied to the Lagrangian dual of our problem.

**Theorem 1** *The proposed iterative procedure for the optimization problem described in ( 1) converges to the unique optimal solution, provided that feasible initial conditions are used and the total KL divergence remains finite for all feasible pseudo-labels.*

The Kullback-Leibler (KL) divergence, $D_{\text{KL}}(p\|q)$, is a well-known convex function in $p$ when $q$ is fixed. The optimization objective function is a sum of convex KL divergence terms. Consequently, the entire objective function is convex. Since the optimization problem consists of minimizing a convex function subject to linear constraints, the problem is a convex optimization problem. Convexity ensures that there is a unique global minimum.

The iterative algorithm begins with feasible initial conditions, where the $f(x_f; \mathbf{w})$ and $f(x_r; \mathbf{w})$ satisfy the constraints ( 2) ( 3) ( 4). These feasible initial conditions guarantee that the optimization process starts in the valid region and remains within this region during the optimization. Because the objective function is convex and the constraints are linear, the iterative procedure will converge to the global optimal solution. Strong duality ensures that the primal and dual solutions will converge to a common point, satisfying both the objective function and the constraints.

The uniqueness of the solution follows from the strict convexity of the KL divergence and the linear constraints. Therefore, the iterative procedure converges to the unique global minimum, as guaranteed by the structure of the problem.

Given this property, we can choose an initialization that is close to the pseudo-probabilities. Starting from a point near the optimal solution significantly reduces the number of iterations required for convergence. This improves computational efficiency by reducing the overall cost of optimization, while still guaranteeing that the solution is optimal.

### 4.1.2 INTEGRATION INTO DEEP MACHINE UNLEARNING

Moreover, our approach can be combined with other unlearning methods. After an initial unlearning phase conducted using existing techniques, our post-processing step can further refine the model's output distribution, ensuring that the unlearning is both comprehensive and efficient.

### 4.2 PROOF FOR OPTIMIZATION STRATEGY

We now provide a detailed mathematical proof to establish the connection between the optimization strategy for model unlearning and the adaptive post-learning method, using Lagrangian duality and iterative coordinate ascent.

### 4.2.1 LAGRANGIAN DUAL FORMULATION

Consider the optimization problem where the goal is to find the refined probabilities $f_k(x_f; \mathbf{w})$ and $f_k(x_r; \mathbf{w})$ for the forget and retain sets, respectively, to minimize the objective function (1).

The objective is to adjust the model outputs $f(x_f; \mathbf{w})$ and $f(x_r; \mathbf{w})$ such that the pseudo-labels for the forget set $f(x_f; \mathbf{w})$ obscure the model's learned associations while ensuring that the retain set pseudo-labels $f(x_r; \mathbf{w})$ are aligned with the model's original predictions.

To handle the class distribution constraints, we introduce dual variables $\alpha_k$ associated with the class distribution constraint for each class $k$ and define the Lagrangian as follows:

$$
\begin{aligned}
\mathcal{L}(f(x_f; \mathbf{w}), f(x_r; \mathbf{w}), \alpha) = \sum_{f=1}^{N_f} D_{\text{KL}}(\hat{p}_f \parallel f(x_f; \mathbf{w})) + \lambda \sum_{r=1}^{N_r} D_{\text{KL}}(\hat{p}_r \parallel f(x_r; \mathbf{w})) \\
+ \sum_k \alpha_k \left( \sum_{f=1}^{N_f} f_k(x_f; \mathbf{w}) + \sum_{r=1}^{N_r} f_k(x_r; \mathbf{w}) - M_k \right)
\end{aligned}
\tag{5}
$$

The Lagrangian formulation allows us to handle the constraints directly by incorporating them into the objective function using the dual variables $\alpha_k$. Given the convexity of the objective function, strong duality holds, meaning that the optimal solution can be found by solving the Lagrangian dual problem.

### 4.2.2 SOLUTION VIA COORDINATE ASCENT

The coordinate ascent method can now be applied to solve the optimization problem. The dual variables $\alpha_k$ are updated iteratively to ensure that the class distribution constraints are satisfied. For each iteration, the primal variables $f(x_f; \mathbf{w})$ and $f(x_r; \mathbf{w})$ are updated to minimize the Lagrangian, followed by updates to the dual variables $\alpha_k$ to satisfy the class constraints.

The primal and dual updates can be written as:

$$
f_k(x_f; \mathbf{w}) = A_{f,k} e^{-\frac{w_k + \alpha_k}{w_k}}, \quad f_k(x_r; \mathbf{w}) = A_{r,k} e^{-\frac{w_k + \alpha_k}{w_k}}
\tag{6}
$$

where $A_{f,k}$ and $A_{r,k}$ are the initial probabilities.

The dual variable update follows:

$$\alpha_k^{(t+1)} = \alpha_k^{(t)} + \eta \left( \sum_{f=1}^{N_f} f_k(x_f; \mathbf{w}) + \sum_{r=1}^{N_r} f_k(x_r; \mathbf{w}) - M_k \right) \qquad (7)$$

where $\eta$ is the step size.

### 4.3 CHANGE THE WEIGHTS

After updating the probabilities, we adjust the model's weights accordingly by using the KL divergence as the loss function to calculate the loss.

## 5 EXPERIMENT

### 5.1 DATASETS AND METRICS

In this study, we employ two distinct datasets that were also used in prior research: CIFAR-10 and Lacuna-10. Lacuna-10 is a curated dataset formed by selecting data from 10 distinct classes, randomly chosen from the extensive VGG-Face2 dataset(Cao et al., 2018). These selected classes each have a minimum of 500 samples, with the data further segmented into 400 training and 100 testing images per class. Lacuna-100 expands on this concept by selecting 100 classes with the same criteria. Our evaluation metric focuses on the model's accuracy, specifically assessing its performance on both the forget set and the retain set to evaluate memory retention. Additionally, we measure the model's resistance to membership inference attacks for the privacy task.

### 5.2 IMPLEMENTATION DETAILS

To facilitate a comprehensive comparison with the performance of other models, we follow the setup in (Kurmanji et al., 2024). We establish two experimental conditions: small-scale and large-scale. The small-scale setting, referred to as CIFAR-5/Lacuna-5, involves a subset of 5 classes from each dataset, comprising 100 training, 25 validation, and 100 testing samples per class. Notably, the forget set includes 25 samples from the initial class, accounting for 5% of the dataset. Conversely, the large-scale setting encompasses all classes from both CIFAR-10 and Lacuna-10, providing a broader spectrum for analysis. In the large-scale scenario, we will explore both class unlearning and selective unlearning. For class unlearning, we define the forget set as the entirety of the training set for class 5, which constitutes 10% of the data. In the selective unlearning scenario, we aim to forget 100 examples from class 5, representing 0.25% of CIFAR-10 and 2% of Lacuna-10.

To align with precedents in the field, our experiments will be conducted using two established architectures: ResNet-18 and ALL-CNN (Springenberg et al., 2014). The baseline model will be pretrained on CIFAR-100 and Lacuna-100 datasets for initial weight setting. Additionally, $\lambda$ will be set to a default value of 1 in the following experiments.

### 5.3 BASELINE

Our approach is benchmarked against the latest state-of-the-art methods and established baselines to highlight its efficacy: **Retrain**: This involves retraining the original model solely on the retain set $\mathcal{D}_r$, considered the gold standard. However, this method is typically deemed impractical for real-world applications. **Original**: TThe baseline model trained on the complete dataset $\mathcal{D}$, without any modifications for data forgetting. **Finetuning**: The original model is fine-tuned on the retain set $\mathcal{D}_r$, incorporating no specific forgetting mechanism. **NegGrad+** (Kodge et al., 2023): An innovative method that applies gradient ascent to the forget set and gradient descent to the retain set over 500 iterations. **Fisher Forgetting** (Golatkar et al., 2020a): Adjusts the model's weights to effectively "unlearn" the data meant to be forgotten, simulating a scenario where the model was never exposed to this data. **NTK Forgetting** (Doan et al., 2021): Employs novel techniques like PCA-OGD to minimize forgetting by orthogonally projecting onto principal directions, preserving

Table 1: Unlearning results with ResNet-18 for the bias removal task. Our method achieves higher forget rates while preserving overall model performance. "PPU w/ uniform" indicates that pseudo-probabilities are set to a uniform distribution, while "PPU w/ random" refers to pseudo-probabilities following a random distribution.

| Model | CIFAR-5 | | | Lacuna-5 | | |
|---|---|---|---|---|---|---|
| | Test error (↓) | Retain error (↓) | Forget error (↑) | Test error (↓) | Retain error (↓) | Forget error (↑) |
| Retrain | 24.90 | **0.00** | 28.80 | 5.80 | **0.00** | 4.80 |
| Original | 24.20 | **0.00** | 0.00 | 5.70 | **0.00** | 0.00 |
| Finetune | 24.30 | **0.00** | 0.00 | 5.60 | **0.00** | 0.00 |
| Fisher | 31.60 | 14.00 | 4.80 | 6.70 | 14.00 | 6.40 |
| NTK | 24.40 | **0.00** | 22.40 | 5.60 | **0.00** | 0.00 |
| NegGrad+ | 25.50 | **0.00** | 41.3 | 6.10 | **0.00** | 1.30 |
| CF-k | 22.60 | **0.00** | 0.00 | 5.80 | **0.00** | 0.00 |
| EU-k | 23.50 | **0.00** | 10.70 | 5.90 | **0.00** | 0.00 |
| Bad-T | 22.73 | 5.12 | 8.00 | 5.00 | 8.64 | 0.14 |
| SCRUB | 24.20 | **0.00** | 40.80 | 6.20 | **0.00** | 24.80 |
| **PPU w/ random** | **22.00** | **0.00** | **80.00** | **2.20** | **0.00** | 64.00 |
| **PPU w/ uniform** | 27.00 | 0.21 | **80.00** | 2.80 | 0.42 | **68.00** |

Table 2: Seletive unlearning results with ALL-CNN for the bias removal task. Our method achieves higher forget rates while preserving overall model performance. "PPU w/ uniform" indicates that pseudo-probabilities are set to a uniform distribution, while "PPU w/ random" refers to pseudo-probabilities following a random distribution.

| Model | CIFAR-10 | | | Lacuna-10 | | |
|---|---|---|---|---|---|---|
| | Test error (↓) | Retain error (↓) | Forget error (↑) | Test error (↓) | Retain error (↓) | Forget error (↑) |
| Retrain | 16.71 | **0.00** | 25.67 | 1.60 | **0.00** | 0.67 |
| Original | **16.43** | **0.00** | 0.00 | 1.53 | **0.00** | 0.00 |
| Finetune | 16.50 | **0.00** | 0.00 | 1.43 | **0.00** | 0.00 |
| Fisher | 21.39 | 4.00 | 13.00 | 1.87 | 0.01 | 0.00 |
| NegGrad+ | 21.36 | 3.23 | 45.33 | 2.77 | 0.40 | 8.67 |
| CF-k | 16.29 | **0.00** | 0.00 | 1.53 | **0.00** | 0.00 |
| EU-k | 17.62 | 0.11 | 0.33 | 1.83 | **0.00** | 0.00 |
| Bad-T | 22.43 | 10.13 | 1.67 | 4.90 | 1.34 | 0.67 |
| SCRUB | 16.55 | **0.00** | 20.33 | 2.07 | **0.00** | 1.67 |
| **PPU w/ random** | 17.00 | **0.00** | 86.00 | **2.20** | **0.00** | 64.00 |
| **PPU w/ uniform** | 16.60 | **0.00** | **95.00** | 2.80 | 0.42 | **68.00** |

data structure integrity. **CF-k, EU-k** (Goel et al., 2022): These methods focus on the model's last k layers. "Exact-unlearning" (EU-k) re-trains these layers from scratch, while "Catastrophic Forgetting" (CF-k) fine-tunes them on the retain set $\mathcal{D}_r$. **SCRUB**(Kurmanji et al., 2024): Introduces a novel training objective and has demonstrated superior performance in prior metrics.

## 5.4 REMOVE BIAS

In addressing bias removal, our goal is to maximize the forget set error. Thus, instead of performing optimizations, we can directly modify the forget set probabilities to reflect pseudo probabilities. Additionally, we experimented with various distributions for the pseudo-probabilities, including uniform and random distributions.Specifically, the random distribution will be generated by applying the softmax function to randomly generated numbers. In general, the uniform distribution tends to perform better in terms of forget error, but it often leads to worse results in test error and retain error. As shown in Table 1, our method PPU comes with a much higher forget error (60-80 percent higher), which is the desired outcome in unlearning scenarios. Table 2 also demonstrates that PPU achieves better performance in forget error for selective unlearning in larger models. Based on the forget error metric, our method appears to be the most successful at unlearning, achieving the desired outcome of complete forgetfulness without severely compromising the performance of the data that should be retained. Additionally, our method exhibits the lowest test error, demonstrating that the model's performance and generalizability are well-preserved even after applying our unlearning technique. The results demonstrate that using pseudo-probabilities is effective for bias removal.

Table 3: Unlearning results with ALL-CNN for the privacy protection task.

| Model | CIFAR-10 | | | Lacuna-10 | | |
|---|---|---|---|---|---|---|
| | Test error | Retain error | Forget error | Test error | Retain error | Forget error |
| Retrain | 16.71 | 0.00 | 26.67 | 1.50 | 0.00 | 0.33 |
| Original | 16.71 | 0.00 | 0.00 | 1.57 | 0.00 | 0.00 |
| Finetune | 16.86 | 0.00 | 0.00 | 1.40 | 0.00 | 0.00 |
| NegGrad+ | 21.65 | 4.54 | 47.00 | 3.60 | 0.87 | 14.33 |
| CF-k | 16.82 | 0.00 | 0.00 | 1.57 | 0.00 | 0.00 |
| EU-k | 18.44 | 0.32 | 0.33 | 3.90 | 0.76 | 0.00 |
| Bad-T | 22.43 | 10.13 | 1.67 | 4.90 | 0.67 | 1.34 |
| SCRUB | 17.01 | 0.00 | 33.00 | 1.67 | 0.00 | 0.00 |
| SCRUB+R | 16.88 | 0.00 | 26.33 | 1.67 | 0.00 | 0.00 |
| **PPU** | 18.05 | 0.00 | 25.35 | 1.05 | 0.00 | 0.05 |

## 5.5 PROTECT PRIVACY

To protect privacy, our goal is to ensure that the forget error remains close to that of retraining. For membership inference attacks, we adopt the approach outlined by Kurmanji et al.(Kurmanji et al., 2024). Specifically, we train a binary classifier (the "attacker") using the losses of the unlearned model on both the forget and test examples, with the objective of classifying instances as either "in" (forget) or "out" (test). The attacker then predicts labels for held-out losses—losses that were not used during training—balanced between the forget and test sets. A successful defense is indicated by an attacker accuracy of 50%, signifying that the attacker is unable to distinguish between the two sets, demonstrating the effectiveness of the unlearning method.

To preserve privacy, we monitor both the training and retain accuracy at each epoch. As shown in Figure 2, the experiment on selective unlearning with ALL-CNN on CIFAR-10 reveals that the forget error gradually increases during training. Therefore, checkpoints are saved at each epoch, and the model closest to the original is selected. In this setup, pseudo-probabilities are initialized using a uniform distribution. According to Table 3, PPU's forget error is very close to that of retraining, particularly in the Lacuna-10 experiment, where it is the closest match. In the membership inference attack experiment, shown in Table 4, PPU consistently achieves nearly 50% accuracy, indicating strong privacy preservation. This demonstrates that, with the refinement of pseudo-probabilities, the model can maintain the original distribution while effectively forgetting the designated forget set.

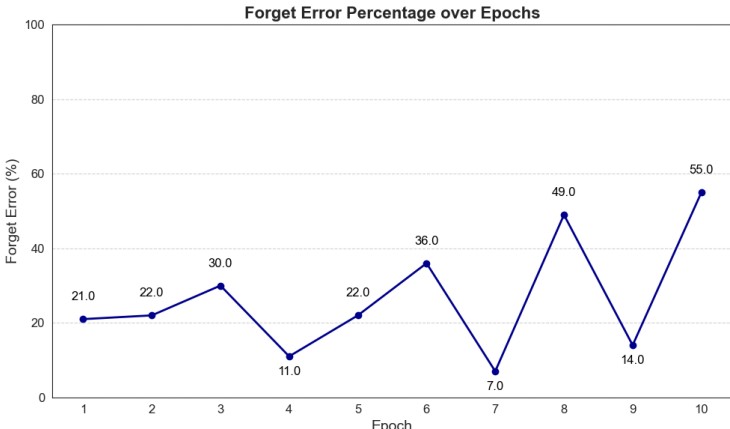

Figure 2: Forget set error on selective unlearning with ALL-CNN on CIFAR-10

## 5.6 COMPUTATIONAL EFFICIENCY

We compare the time required for SCRUB (Kurmanji et al., 2024), retraining, and our method, with all experiments conducted on an NVIDIA RTX-4090. Time is recorded over 5 runs, and we report both the mean and the standard error. In Figure 3, we present the time required for the bias removal tasks using the ResNet-18 model and selective unlearning using ALL-CNN. Compared to other

Table 4: Membership inference attack results with ResNet-18 and ALL-CNN in large-scale unlearning. The closer the result is to 50%, the better the performance.

| | ResNet | | | | ALL-CNN | | | |
| | Class | | Selective | | Class | | Selective | |
| Model | mean | std | mean | std | mean | std | mean | std |
|---|---|---|---|---|---|---|---|---|
| Retrain | 49.33 | 1.67 | 54.00 | 1.63 | 55.00 | 4.00 | 48.73 | 0.24 |
| Original | 71.10 | 0.67 | 65.33 | 0.47 | 66.50 | 0.50 | 71.40 | 0.70 |
| Finetune | 75.57 | 0.69 | 64.00 | 0.82 | 68.00 | 1.00 | 74.97 | 1.27 |
| NegGrad+ | 69.57 | 1.19 | 66.67 | 1.70 | 72.00 | 0.00 | 70.03 | 1.92 |
| CF-k | 75.73 | 0.34 | 65.00 | 0.00 | 69.00 | 2.00 | 72.93 | 1.06 |
| EU-k | 54.20 | 2.27 | 53.00 | 3.27 | 66.50 | 3.50 | 51.60 | 1.22 |
| Bad-T | 54.00 | 1.10 | 59.67 | 4.19 | 63.4 | 1.2 | 77.67 | 4.11 |
| SCRUB | 52.20 | 1.71 | 78.00 | 2.45 | 52.00 | 0.00 | 54.30 | 2.24 |
| SCRUB+R | 52.20 | 1.71 | 58.67 | 1.89 | **52.00** | 0.00 | 54.30 | 2.24 |
| **PPU** | **51.00** | 1.05 | **58.00** | 0.93 | 54.00 | 0.70 | **50.00** | 0.40 |

methods, PPU significantly reduces computation time, cutting it to less than half of what is required by SCRUB. The results further emphasize the high effectiveness of the optimization approach and the use of pseudo-probabilities to fine-tune the model weights.

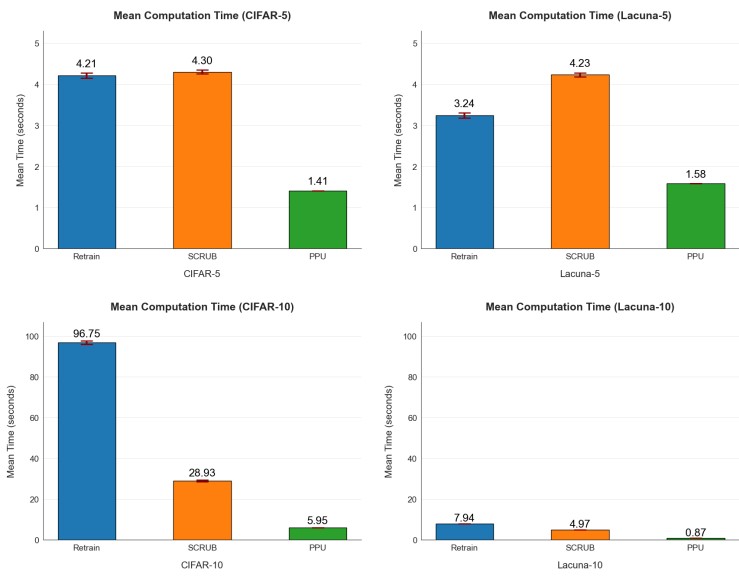

Figure 3: Time needed for the unlearning method (measured over 5 runs)

## 5.7 ADAPTIVE UNLEARNING

Our method can also be applied as post-processing after unlearning methods to enhance their results. PPU can be considered a plug-in that is compatible with nearly all existing methods. In our experiments, we built on SCRUB (Kurmanji et al., 2024) and applied our method afterward. For the bias removal task, this approach improves forget error by more than 50%, with less than a 0.5% decrease in retain error. Detailed results can be found in Appendix A.1. In addition to SCRUB, we applied our method after fine-tuning on CIFAR-10 with a pretrained ResNet, achieving a 2.5% retain error and a 60% forget error. In comparison, the original fine-tuning method achieved only a 2% retain error and a 16% forget error.

## 6 ABLATION STUDY

In the optimization objective function ( 1), the value of $\lambda$ was set to 1 in all previous experiments. Here, we explore the impact of varying $\lambda$ on the retain and forget errors in a small-scale unlearn-

Table 5: The retain error and forget error with varying $\lambda$ values were evaluated in a small-scale unlearning experiment on CIFAR-5 using ResNet.

| Model | $\lambda = 1$ | | $\lambda = 2$ | | $\lambda = 3$ | | $\lambda = 4$ | |
| | Retain error | Forget error | Retain error | Forget error | Retain error | Forget error | Retain error | Forget error |
|---|---|---|---|---|---|---|---|---|
| **PPU** | 0.21 | 80.00 | 0.00 | 56.00 | 0.00 | 23.00 | 0.00 | 30.00 |

ing experiment on CIFAR-5 with ResNet. As $\lambda$ increases, more weight is assigned to the retain set, resulting in a decrease in retain error from 0.21% to 0%. However, this reduction comes at a significant cost to the forget error.

To investigate our method in a larger setting, we also conducted an additional experiment on the CIFAR-100 dataset with one class unlearning. Our method demonstrated very good performance. Using the ResNet architecture, SCRUB achieved a forget error of 5.19 and a retain error of 0.00015. In contrast, our method achieved a retrain 031 error of 0.00 and a forget error of 98.25.

## 7 CONCLUSION

This research introduces a novel approach to machine unlearning, presenting an optimization framework that refines predictive probability distributions within deep learning models. Our method excels in striking an optimal balance between forgetting effectiveness and preserving model performance on retained data. Additionally, it demonstrates superior resilience against membership inference attacks. Empirical results across diverse datasets and model architectures, including CIFAR-10 and Lacuna-10 with ResNet and ALL-CNN, highlight the superiority of our approach over existing state-of-the-art methods.

Furthermore, the operational flexibility, theoretical insights, and high computational efficiency of our approach provide a solid foundation for further developments. However, we acknowledge certain limitations. Our current method is limited to addressing unlearning in classification tasks and may encounter convergence issues during the optimization process. Additionally, the approach is restricted to supervised learning settings and does not extend to unsupervised tasks at this stage. Future work will focus on extending the method to various models, including large language models, and broadening its applicability beyond classification tasks.

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

# A APPENDIX

## A.1 MORE RESULTS FOR ADAPTIVE UNLEARNING

Here, we present additional results for our method, applied as a post-processing step after SCRUB. For the bias removal task, our method significantly improves the forget error while having minimal impact on the model's original performance. The results are incorporated in Table 6, Table 7, Table 8, Table 9, and Table 10.

Table 6: Unlearning results with ALL-CNN. Our method gets top performance in forget with little influence on model performance (retain error).

| Model | CIFAR-5 | | Lacuna-5 | |
| --- | --- | --- | --- | --- |
| | Retain error (↓) | Forget error (↑) | Retain error (↓) | Forget error (↑) |
| Retrain | 0.13 | 28.80 | **0.00** | 4.67 |
| Original | 0.17 | 0.00 | **0.00** | 0.00 |
| Finetune | 0.04 | 0.00 | 6.63 | 19.33 |
| Fisher | 31.83 | 15.20 | 51.09 | 39.33 |
| NTK | 0.17 | 13.6 | **0.00** | 3.33 |
| NegGrad+ | 0.56 | 36.00 | 0.14 | 12.00 |
| CF-k | **0.00** | 0.00 | **0.00** | 0.00 |
| EU-k | 3.23 | 8.00 | **0.00** | 0.00 |
| Bad-T | 9.68 | 10.67 | 2.32 | 0.00 |
| SCRUB | 0.08 | 40.80 | **0.00** | 25.33 |
| **SCRUB + PPU** | 1.05 | **68.00** | 1.47 | **78.00** |

Table 7: Class unlearning results with ResNet. Our method gets top performance in forget with little influence on model performance (retain error).

| Model | CIFAR-10 | | Lacuna-10 | |
| --- | --- | --- | --- | --- |
| | Retain error (↓) | Forget error (↑) | Retain error (↓) | Forget error (↑) |
| Retrain | **0.00** | 100.00 | **0.00** | 99.75 |
| Original | **0.00** | 0.00 | **0.00** | 0.00 |
| Finetune | **0.00** | 0.00 | **0.00** | 0.00 |
| Fisher | 2.45 | **100.00** | **0.00** | **100.00** |
| NegGrad+ | 1.74 | 91.26 | **0.00** | 14.90 |
| CF-k | **0.00** | 0.03 | **0.00** | 0.00 |
| EU-k | **0.00** | 98.79 | 0.01 | 4.06 |
| Bad-T | 11.34 | 94.67 | 1.06 | 67.60 |
| SCRUB | 0.51 | **100.00** | 0.28 | **100.00** |
| **SCRUB+PPU** | 2.48 | **100.00** | **0.00** | **100.00** |

Table 8: Class unlearning results with ALL-CNN. Our method gets top performance in forget with little influence on model performance (retain error).

| Model | CIFAR-10 | | Lacuna-10 | |
| --- | --- | --- | --- | --- |
| | Retain error (↓) | Forget error (↑) | Retain error (↓) | Forget error (↑) |
| Retrain | **0.00** | 100.00 | **0.00** | 100.00 |
| Original | **0.00** | 0.00 | **0.00** | 0.00 |
| Finetune | **0.00** | 0.00 | **0.00** | 0.00 |
| Fisher | 3.66 | 99.00 | **0.00** | 89.00 |
| NegGrad+ | 0.58 | 87.22 | **0.00** | 6.56 |
| CF-k | **0.00** | 0.00 | **0.00** | 0.00 |
| EU-k | 0.13 | **100.00** | **0.00** | 77.19 |
| Bad-T | 5.84 | 81.93 | 0.37 | 38.65 |
| SCRUB | 0.12 | **100.00** | **0.00** | **100.00** |
| **SCRUB+PPU** | 0.20 | **100.00** | **0.00** | **100.00** |

Table 9: Selective unlearning results with ResNet. Our method gets top performance in forget with little influence on model performance (retain error).

| Model | CIFAR-10 | | Lacuna-10 | |
| --- | --- | --- | --- | --- |
| | Retain error (↓) | Forget error (↑) | Retain error (↓) | Forget error (↑) |
| Retrain | **0.00** | 29.67 | **0.00** | 1.0 |
| Original | **0.00** | 0.00 | **0.00** | 0.00 |
| Finetune | **0.00** | 0.00 | **0.00** | 0.00 |
| Fisher | 2.88 | 3.00 | **0.00** | 0.00 |
| NegGrad+ | 4.10 | 53.70 | 0.90 | 13.00 |
| CF-k | **0.00** | 0.00 | **0.00** | 0.00 |
| EU-k | 0.40 | 23.70 | 0.00 | 0.00 |
| Bad-T | 14.53 | 34.67 | 3.26 | 0.33 |
| SCRUB | **0.00** | 70.33 | **0.00** | 4.67 |
| **SCRUB+PPU** | 0.01 | **100.00** | 5.39 | **100.00** |

Table 10: Selective unlearning results with ALL-CNN. Our method gets top performance in forget with little influence on model performance (retain error).

| Model | CIFAR-10 | | Lacuna-10 | |
|---|---|---|---|---|
| | Retain error ($\downarrow$) | Forget error ($\uparrow$) | Retain error ($\downarrow$) | Forget error ($\uparrow$) |
| Retrain | **0.00** | 25.67 | **0.00** | 0.67 |
| Original | **0.00** | 0.00 | **0.00** | 0.00 |
| Finetune | **0.00** | 0.00 | **0.00** | 0.00 |
| Fisher | 4.00 | 13.00 | 0.01 | 0.00 |
| NegGrad+ | 3.23 | 45.33 | 0.40 | 8.67 |
| CF-k | **0.00** | 0.00 | **0.00** | 0.00 |
| EU-k | 0.11 | 0.33 | 0.00 | 0.00 |
| Bad-T | 10.13 | 1.67 | 1.34 | 0.67 |
| SCRUB | **0.00** | 29.33 | **0.00** | 1.67 |
| **SCRUB+PPU** | **0.00** | **100.00** | 0.00 | **88.12** |

