# OpenReview forum: "Pseudo-Probability Unlearning: Towards Efficient and Privacy-Preserving Machine Unlearning"
_ICLR.cc/2025/Conference — ICLR 2025 Conference Withdrawn Submission_

### Official Review · Reviewer_8GnQ · 2024-10-23

**Soundness:** 2
**Presentation:** 1
**Contribution:** 1
**Rating:** 3
**Confidence:** 5

**Summary:**

The paper tries to solve the efficiency and privacy leakage of machine unlearning. The problem and topic is interesting and hot. I have some comments as follows.

Strengths: 1. The experiments and evaluation are sufficient.

Weaknesses:
1. The problem statement is not clear. In the introduction and abstract, the authors claimed that they aim to solve the efficiency and privacy leakage of machine unlearning. However, in the section 3, the problem definition section, it has not mentioned the definition related to the efficiency and privacy leakage problems in machine unlearning.

2. Regarding the privacy leakage in machine unlearning, the authors ignored the common attacks that compare the model difference before and after unlearning to implement inference attacks. To the reviewer's understanding, the proposed method is infeasible for these attacks in machine unlearning because the original model performs well on erased samples but now randomly predicts them, which is a huge difference.

3. The paper needs heavy proofreading. There are many typos, for example "Figure ??" in page 2.

**Strengths:**

1. The experiments and evaluation are sufficient.

**Weaknesses:**

1. The problem statement is not clear. In the introduction and abstract, the authors claimed that they aim to solve the efficiency and privacy leakage of machine unlearning. However, in the section 3, the problem definition section, it has not mentioned the definition related to the efficiency and privacy leakage problems in machine unlearning.

2. Regarding the privacy leakage in machine unlearning, the authors ignored the common attacks that compare the model difference before and after unlearning to implement inference attacks. To the reviewer's understanding, the proposed method is infeasible for these attacks in machine unlearning because the original model performs well on erased samples but now randomly predicts them, which is a huge difference.

3. The paper needs heavy proofreading. There are many typos, for example "Figure ??" in page 2.

**Questions:**

No additional questions.

---

> ### Author Response · Authors · 2024-11-25
> **Response to Reviewer 8GnQ**
>
> Thank you for recognizing the extensive experimentation in our work. We appreciate your suggestions, and our responses are as follows:
>
> **(1) Efficiency and Privacy Leakage:** We evaluate both efficiency and privacy in Sections 5.5 and 5.6. Privacy is primarily assessed using the *membership inference attack*, while efficiency is measured by the* total computation time* required. The results of the membership inference attack are presented in Table 4. We will clearly define these in the revised version.
>
> **(2) Random Distribution vs. Membership Inference Attack:** The random distributions are proposed to achieve a high forget error on the forget set without requiring optimization to maximize the forget error. This approach is cost-efficient and does not require access to the forget set's labels.
> Membership inference attack is *not* mitigated by the uniform distribution itself but by applying the optimization goal upon the uniform distribution to *minimize the discrepancy* between the retained and unlearned models, as shown in Equation 1. This step ensures that the unlearned model remains close to the retained model, improving performance on the retained set and reducing the risk of membership inference attacks on the forget set. Table 4 shows the effectiveness of our approach in mitigating membership inference compared with baselines.
>
> **(3) Typos:** We appreciate the reviewer for pointing out typos in the original submission. The reference to "Figure ??" originally refers to Figure 1. We have addressed and corrected this, along with several other typos, in the rebuttal version.

---

> > ### Comment · Reviewer_8GnQ · 2024-11-26
> > **Official Comment by Reviewer 8GnQ**
> >
> > Thank you for the response. I still think the paper needs to be heavily polished and I will maintain my rating.

---

### Official Review · Reviewer_P32Y · 2024-11-02

**Soundness:** 2
**Presentation:** 2
**Contribution:** 2
**Rating:** 3
**Confidence:** 4

**Summary:**

This research studies methods for achieving unlearning, which involves forgetting information about specific data from a trained model. In particular, it points out that with existing unlearning methods, it is possible to identify whether or not a piece of data has been unlearned by using a membership inference attack and that this can lead to privacy violations. This research proposes a method that enables unlearning while preventing such privacy violations. Specifically, the authors propose a method that not only makes it impossible to classify the data to be forgotten but also makes the output probability of the final layer when classifying the data to be forgotten to match a uniform distribution or a distribution with random probabilities for each class. They argue that this prevents privacy violations associated with unlearning. They also consider optimization methods for implementing such a method.

**Strengths:**

The perspective on privacy violations of unlearning through membership inference is important, and this paper introduces this perspective as a new contribution.

**Weaknesses:**

It is not sufficiently discussed whether the proposed method can indeed prevent privacy violations of unlearning through membership inference. In addition, there is no experimental comparison with other methods from this perspective.

**Questions:**

The motivation for the proposed method in the introduction was the privacy violation in unlearning through membership inference. However, in the experiment, only test, retain and forget errors were evaluated. There was no experimental evaluation of privacy violations through membership inference. It is necessary to experimentally evaluate whether the proposed method is more robust against privacy violations through membership inference than other methods.

The proposed method claims that if the probability output of the final layer is uniform or random, it can avoid membership inference. Is this true? It is unlikely that the probability output of the final layer is uniform by chance. Could it be evidence that the probability output of the final layer was the target of unlearning?

The same applies if the probability output of the final layer is random. If such randomness does not appear in any other test data, it is possible to infer that it is artificial randomness. To show that the data to be forgotten is not identified by membership inference, it may be necessary to show, for example, using statistical testing, that the probability output of the final layer for the data contained in the test data is statistically indistinguishable from the probability output of the data to be forgotten.

---

> ### Author Response · Authors · 2024-11-25
> **Response to Reviewer P32Y**
>
> Thank you for recognizing our work's contribution to privacy on machine unlearning. We appreciate your suggestions, and our responses are below:
>
> **(1) Membership Inference Attack:** The evaluation of the membership inference attack is detailed in Section 5.5, titled "Protect Privacy." The results, which compare our method to others, are presented in Table 4. In this table, our proposed method, PPU, demonstrates superior performance compared to alternative approaches, particularly in the ResNet large-scale class and selective unlearning tasks, as well as the ALL-CNN selective unlearning task.
>
> **(2) Uniform/Random Distribution vs. Membership Inference Attack:** The uniform and random distributions are proposed to achieve a high forget error on the forget set without requiring optimization to maximize the forget error. This approach is cost-efficient and does not require access to the forget set's labels.
> Membership inference attack is *not* mitigated by the uniform distribution itself but by applying the optimization goal upon the uniform distribution to *minimize the discrepancy* between the retained and unlearned models, as shown in Equation 1. This step ensures that the unlearned model remains close to the retained model, improving performance on the retained set and reducing the risk of membership inference attacks on the forget set.

---

> > ### Comment · Reviewer_P32Y · 2024-11-29
> > **Reponse to the authors**
> >
> > Thank you for your reply. Some of the concerns are resolved, while concerns about artificial randomness are not resolved, and I'd like to keep the current score.

---

### Official Review · Reviewer_nd1o · 2024-11-04

**Soundness:** 2
**Presentation:** 2
**Contribution:** 2
**Rating:** 5
**Confidence:** 4

**Summary:**

The paper proposes Pseodo-Probability Unlearning (PPU), a novel method that enables models to forget data efficiently and in a privacy-preserving manner. PPU replaces the final-layer output probabilities of the neural network with pseudo-probabilities for the data to be forgotten. The pseudo-probabilities are initialized from some distributions (e.g., a uniform distribution) and are further refined to maintain performance on the remaining data. Extensive experiments demonstrate the efficiency of PPU while preventing privacy leakage.

**Strengths:**

1. The idea of modifying the final-layer output probabilities of the neural network to pseudo-probabilities for machine unlearning is novel.
2. The paper is overall well-structured and generally easy to follow.

**Weaknesses:**

1. Adding an independent paragraph for contributions in the Introduction Section would be better.
2. The method SISA is considered to be exact unlearning.
3. Some typos. "Figure ??" on line 080. $\mathcal D_r$ and $D_{fog}$ on line 139. "The dual variables $\lambda_k$" on line 269. Missing space characters after $\lambda$s on line 478 and line 480.
4. Miss descriptions of Table 3 in the main text.
5. The font size of the text in the figures is too small.
6. Reuse of the notation $f$ for both the model and the forget set.
7. Experiments on larger datasets and networks (e.g., ImageNet) would be better.

**Questions:**

1. Should $\lambda_k$ in Section 4.2.2 be $\alpha_k$?
2. Should "retrain error" on line 485 be "retain error"?
3. What is $w_k$ in Equation 6?
4. The goals in Section 5.4 and Section 5.5 are different. Are the two goals contradictory? Can PPU achieve both of them simultaneously?

---

> ### Author Response · Authors · 2024-11-25
> **Response to Reviewer nd1o**
>
> Thank you for acknowledging our work as a novel approach to machine unlearning. We sincerely appreciate your valuable suggestions, and our responses are provided below:
>
> **(1) Contribution:** We propose a novel machine unlearning framework that focuses on two key tasks: removing bias and protecting privacy. Our approach introduces a new perspective by optimizing the final layer probabilities, a technique not previously explored by existing unlearning methods to the best of our knowledge. Additionally, we provide proof showing that our method converges to a unique optimal solution and can be efficiently solved using coordinate descent. Our method achieves a higher forget error in bias removal tasks while maintaining performance comparable to a retrained model in privacy protection tasks.
>
> **(2) Typos:** Thanks for pointing out that we incorrectly categorized SISA as an approximate unlearning method. It is, in fact, an exact unlearning method, and we will update this accordingly. Additionally, \lambda_k​ should be replaced with \alpha_k​, and the reference on line 485 should be corrected to "retain error" instead of "retrain error." We have addressed and corrected these, along with several other typos, in the rebuttal version.
>
> **(3) Differences between 5.4 and 5.5:** Goals 5.4 and 5.5 have distinct objectives. Goal 5.4 evaluates unlearning performance on the forget set, measured by the forget set error, where higher values indicate better unlearning. Goal 5.5 evaluates privacy leakage risk, measured by membership inference attacks, where lower values are preferred. These goals are achieved simultaneously by assigning pseudo-probabilities to the forget set's logit layer to optimize the forget error. The unlearned model is then refined by minimizing the discrepancy between the retained and unlearned models, making it harder to differentiate the forget set from other data. We acknowledge that the current section titles may be confusing and will revise them to “5.4 Unlearning Performance” and “5.5 Membership Inference Attack” for clarity.
>
>
> **(4) Experiment on Imagenet**
> We evaluated our method on ImageNet-1K with class unlearning for forgetting a single class. For this experiment, we utilized the ResNet-50 v1.5 architecture [4], which was pre-trained on ImageNet-1K and publicly available. Our results demonstrate that our method achieves a forget error of 82% and a retain error of 30%. Furthermore, under a membership inference attack, it achieves an approximate success rate of 60.2%.

---

> > ### Comment · Reviewer_nd1o · 2024-11-26
> >
> > Thank you for the response. I still think the paper needs further revisions and I will maintain my rating.

---

### Official Review · Reviewer_EiQy · 2024-11-10

**Soundness:** 1
**Presentation:** 1
**Contribution:** 1
**Rating:** 1
**Confidence:** 5

**Summary:**

This paper proposes an efficient machine unlearning solution, Privacy-Preserving Unlearning (PPU), that aims to minimize privacy leakage while maintaining model performance. The proposed algorithm is evaluated on two datasets and two deep neural network models, demonstrating its effectiveness in reducing the forget error on forgotten data. The paper provides a comprehensive evaluation of the algorithm and compares it with multiple baseline unlearning methods.

**Strengths:**

- The paper addresses an important problem of efficient and privacy-preserving machine learning and provides a practical solution.
- The evaluation covers two datasets, two models, and multiple baseline machine unlearning algorithms.

**Weaknesses:**

- The proposed PPU algorithm seems to only work for deep neural networks and classification tasks and may not be applicable to other types of models or tasks.
- The optimization goal of the PPU algorithm could be wrong. The algorithm aims to maximize the forget error on forgotten data instead of minimizing the discrepancy between the original and unlearned models.
- In the evaluation section, the retain error and forget error metrics could be further explained to provide more insights into the algorithm's behavior.
- There are many editing issues throughout the paper, such as typos, missing citations, and missing figures.

**Questions:**

1. Why does the proposed algorithm focus on minimizing the forget error on forgotten data instead of minimizing the discrepancy between the retrained and unlearned models?
2. Have you considered scenarios where the model's prediction confidence may not accurately reflect the sensitivity of the data?
3. What are the key factors that influence the trade-off between privacy preservation and model performance in the context of machine unlearning?
4. The proposed algorithm seems to only work for classification tasks and small datasets. How scalable is the proposed algorithm to large-scale datasets and complex models? How easy or difficult to adapt the proposed PPU to more challenging datasets or tasks, for instance ImageNet dataset, regression model, or generative model?
5. Can you provide more insights into the retain error and forget error metrics and how they reflect the unlearning algorithm's performance in different scenarios?
6. I do not see the forgot error comparison in Figure 3, and there is also no Figure 4 to support the claim that "According to Figure 3, PPU’s forget error is very close to that of retraining, particularly in the Lacuna-10 experiment, where it is the closest match. In the membership inference attack experiment, shown in Figure 4, PPU consistently achieves nearly 50% accuracy, indicating strong privacy preservation" in Line 408. Can you please check on this?

---

> ### Author Response · Authors · 2024-11-25
> **Response to Reviewer EiQy**
>
> Thank you for recognizing our work's contribution to privacy on machine unlearning. We appreciate your suggestions, and our responses are below:
>
> **(1) Why focus on Classification tasks?**
>
> Classification is an important and foundational task that has been the focus of many prior unlearning methods [1, 2]. We focus on the classification task to ensure comparability with existing approaches. However, we acknowledge the significance of extending unlearning techniques to other tasks, such as generative tasks, and plan to explore these directions in future work.
>
> **(2) Why maximize the forgotten error instead of minimizing the discrepancy between the retained and unlearned models?**
>
> Thank you for your question. We respectfully note a potential inconsistency in the review, as these two statements conflict. If we understand correctly, the reviewer is asking why our optimization goal is to "maximize the forget error" (listed as the first weakness) rather than "minimize the forget error" (listed as the first question).
> To clarify, we do not adopt an optimization-based approach to maximize the forgotten error. Instead, we modify the forget set's probabilities into pseudo-probabilities, which reduces computational costs while maintaining strong forgotten performance. Forgotten performance, as typically evaluated by the forgotten error in prior work [1, 2, 3], is preserved in our approach.
> Following the application of pseudo-probabilities, our method indeed includes *minimizing the discrepancy* between the retained and unlearned models, as shown in Equation 1. This step ensures that the unlearned model remains close to the retained model, resulting in better performance on the retained set and reduced risk of membership inference attacks on the forget set.
>
> **(3) Experiment on Imagenet**
>
> We evaluated our method on ImageNet-1K with class unlearning for forgetting a single class. For this experiment, we utilized the ResNet-50 v1.5 architecture [4], which was pre-trained on ImageNet-1K and publicly available. Our results demonstrate that our method achieves a forget error of 82% and a retain error of 30%. Furthermore, under a membership inference attack, it achieves an approximate success rate of 60.2%.
>
> **(4) Have you considered scenarios where the model's prediction confidence may not accurately reflect the sensitivity of the data?**
>
> The unlearned sensitive dataset is predefined instead of considering the model’s prediction, as outlined in Section 5.2. We adopt the settings described in Kurmanji et al. (2024). The objective of unlearning methods is to ensure that the model forgets the specified samples, making it behave as if it had never been trained on those samples.
>
> **(5) What are the key factors that influence the trade-off between privacy preservation and model performance in the context of machine unlearning?**
>
> To protect privacy, the model's performance on the forget set should be similar to that of a retrained model. If the forget error on the forget set is excessively high, it becomes easier for membership inference attacks to identify the unlearned examples.
>
> **(6) Forget error and retain error metrics.**
>
> The forget error metric quantifies how well the model has forgotten specified data by calculating the error rate (100% minus accuracy) on the forget set. Similarly, the retain error measures the error rate on the retain set, evaluating how much the model's performance on other datasets is preserved. While the forget error highlights the effectiveness of the unlearning process, the retain error ensures that unlearning does not negatively impact the model's performance on unrelated data.
>
>
> **Typo for figure 3 and figure 4.**
>
> Thanks for pointing out this. In line 408, “Figure 3” should be replaced with “Table 3”. This table demonstrates that the model's performance on the forgotten set is comparable to that of the retrained model. Additionally, the membership inference attack is presented Table 4 instead of Figure 4. We have addressed and corrected these, along with several other typos, in the rebuttal version.
>
>
> [1] Kurmanji, M., Triantafillou, P., Hayes, J., & Triantafillou, E. (2024). Towards unbounded machine unlearning. Advances in neural information processing systems, 36.
>
> [2] Kodge, S., Saha, G., & Roy, K. (2023). Deep Unlearning: Fast and Efficient Training-free Approach to Controlled Forgetting.
>
> [3] Chundawat, V. S., Tarun, A. K., Mandal, M., & Kankanhalli, M. (2023, June). Can bad teaching induce forgetting? unlearning in deep networks using an incompetent teacher. In Proceedings of the AAAI Conference on Artificial Intelligence (Vol. 37, No. 6, pp. 7210-7217).
>
> [4] He, K., Zhang, X., Ren, S., & Sun, J. (2016). Deep residual learning for image recognition. In Proceedings of the IEEE conference on computer vision and pattern recognition (pp. 770-778).

---

### Note · Authors · 2024-12-04

I have read and agree with the venue's withdrawal policy on behalf of myself and my co-authors.